# Bridging Neural and Symbolic Computation: A Learnability Study of RNNs on Counter and Dyck Languages

**Neisarg Dave** [*]                                    NEISARG.DAVE@ADP.COM
*Automatic Data Processing, Inc.*

**Dan Kifer**                                           DUK17@PSU.EDU
*The Pennsylvania State University*

**C. Lee Giles**                                        CLG20@PSU.EDU
*The Pennsylvania State University*

**Ankur Mali**                                          ANKURARJUNMALI@USF.EDU
*University of South Florida*

**Editors:** Leilani H. Gilpin, Eleonora Giunchiglia, Pascal Hitzler, and Emile van Krieken

## Abstract

This work presents a neuro-symbolic analysis of the learnability of Recurrent Neural Networks (RNNs) in classifying structured formal languages—specifically, **counter languages** and **Dyck languages**, which serve as canonical examples of context-free and mildly context-sensitive grammars. While prior studies have highlighted the expressive power of first-order (LSTM) and second-order (O2RNN) architectures within the Chomsky hierarchy, we challenge this perspective by shifting the focus from theoretical expressivity to *practical learnability under finite precision constraints*. Our results suggest that RNNs function more as finite-state machines than stack-based automata when implemented with realistic training regimes and embedding representations. We show that classification performance degrades sharply as structural similarities between positive and negative sequences increase—highlighting a core limitation in the RNN's ability to internalize hierarchical structure without symbolic scaffolding. Interestingly, even simple linear classifiers built on top of RNN-derived embeddings outperform chance, underscoring the hidden representational capacity within learned states. To probe generalization, we train models on input lengths up to 40 and evaluate on lengths extending to 500, using 10 distinct seeds to measure statistical robustness. O2RNNs consistently demonstrate greater stability and generalization compared to LSTMs, particularly under varied initialization strategies. These findings expose the fragility of learned language representations and emphasize the role of architectural bias, initialization, and data sampling in determining what is truly learnable. Ultimately, our study reframes RNN learnability through the lens of *symbolic structure and computational constraints*, advocating for stronger formal criteria when assessing neural models' capacity to reason over structured sequences. We argue that expressivity alone is insufficient—**stability, precision, and symbolic alignment** are essential for true neuro-symbolic generalization.

## 1. Introduction

Recurrent neural networks (RNNs) are experiencing a resurgence, spurring significant research aimed at establishing theoretical bounds on their expressivity. As natural neural

---

[*] Work done while at The Pennsylvania State University

analogs to state machines described by the Chomsky hierarchy, RNNs offer a robust framework for examining learnability, stability, and generalization—core aspects for advancing the development of memory-augmented models.

While conventional RNN architectures typically approximate finite state automata, LSTMs have shown the capacity to learn and generalize non-regular grammars, including counter languages and Dyck languages. These non-regular grammars demand a state machine enhanced with a memory component, and LSTM cell states have been demonstrated to possess sufficient expressivity to mimic dynamic counters. However, understanding whether these dynamics are stable and reliably learnable is crucial, particularly as the stability of learned fixed points directly impacts the generalization and reliability of these networks.

In this work, we extend the investigation into the expressiveness of RNNs by focusing on the empirical evidence for learnability and generalization in complex languages, with specific attention to Dyck and counter languages. Our analysis reveals the following key insights: (1) Theoretical expressivity of LSTM does not necessarily translate to practical learnability on dyck and counter languages; (2) The stability of LSTM feature encodings is heavily influenced by the precision of the network's internal dynamics; (3) Higher Order connections (O2RNN) demonstrate consistant performance across different training strategies and random seed; (4) LSTM's ability to perform dynamic counting is closely tied to the stability of its cell state, which relies on the fixed points of the tanh activation function; and (5) the choice of initialization strategy significantly influences the stability of fixed points in RNNs.

Our analysis builds upon results from Omlin and Giles (1996) regarding the behavior of the *sigmoid* activation function, extending this understanding to the fixed points of the *tanh* function used in LSTM cell and hidden state updates. Drawing from parallels noted by Merrill et al. (2020) between LSTMs and counter machines, we show that while LSTM cell states exhibit the expressivity needed for counting, this capability is not reliably captured in the hidden state. As a result, when the difference between successive hidden states falls below the precision threshold of the decoder, the classifier can no longer accurately represent the counter, leading to generalization failure. Additionally, we explore how input and forget gates within the LSTM clear the counter dynamics as state changes accumulate, resulting in an eventual collapse of dynamic behavior.

Further, we extend our exploration to analyze the learnability of classification layers when the encoding RNN is initialized randomly and not trained. This setup allows us to assess the extent of instability induced by the collapse of counter dynamics in the LSTM cell state and the role of numeric precision in the hidden state that supports the classification layer's performance.

It is crucial to recognize that most prior studies demonstrating the learnability of RNNs on counter languages such as $a^n b^n$, $a^n b^n c^n$, and $a^n b^n c^n d^n$ have overlooked the significance of topological distance between positive and negative samples. Such sampling considerations are vital for a thorough understanding of RNN trainability. To address this gap, we incorporate three sampling strategies with varying levels of topological proximity between positive and negative samples, thereby challenging the RNNs to genuinely learn the counting mechanism.

By focusing on stability and fixed-point dynamics, our work offers a plausible lens through which the learnability of complex grammars in recurrent architectures can be bet-

ter understood. We argue that stability, as characterized by the persistence of fixed points, is a critical factor in determining whether these models can generalize and reliably encode non-regular languages, shedding light on the inherent limitations and potentials of RNNs in such tasks.

## 2. Related Work

The relationship between Recurrent Neural Networks (RNNs), automata theory, and formal methods has been a focal point in understanding the computational power and limitations of neural architectures. Early studies have shown that RNNs can approximate the behavior of various automata and formal language classes, providing insights into their expressivity and learnability. Giles and Omlin (1993) were one of the first to demonstrate that RNNs are capable of learning finite automata. Expanding on this, Omlin and Giles (1992b) showed that second-order recurrent networks, which include multiplicative interactions between inputs and hidden states, are superior state approximators compared to standard first-order RNNs.

The analysis of RNNs' functional capacity continued with Omlin and Giles (1996); Mali et al. (2023), who investigated the discriminant functions underlying first-order and second-order RNNs. Their results provided a deeper understanding of how these architectures utilize hidden state dynamics to implement decision boundaries and process temporal patterns. Meanwhile, the theoretical limits of RNNs were formalized by Siegelmann and Sontag (1992), who proved that RNNs are Turing Complete when equipped with infinite precision. This result implies that RNNs, in principle, can simulate any computable function, positioning them as universal function approximators.

Building on these foundational insights, recent research has aimed to identify the practical scenarios under which RNNs can achieve such theoretical expressiveness. Mali et al. (2023) extended Turing completeness results to a second-order RNN, demonstrating that it can achieve Turing completeness in bounded time. This shift towards practical expressivity has opened new avenues for applying RNNs to complex language tasks. Moving towards specific language modeling tasks, Gers and Schmidhuber (2001) explored how Long Short-Term Memory (LSTM) networks can learn context-free and context-sensitive grammars, such as $a^n b^n$ and $a^n b^n c^n$. Their results showed that LSTMs could successfully learn these patterns, albeit with limitations in scaling to larger sequence lengths. Extending these findings, Merrill et al. (2020) established a formal hierarchy categorizing RNN variants based on their expressivity, placing LSTMs in a higher class due to their ability to simulate counter machines. Suzgun et al. (2019a) analyzed the ability of LSTMs to learn the Dyck-1 language, which models balanced parentheses, and found that while a single LSTM neuron could learn Dyck-1, it failed to generalize to Dyck-2, a more complex language with nested dependencies. Their follow-up work Suzgun et al. (2019b) studied generalization on $a^n b^n$, $a^n b^n c^n$, and $a^n b^n c^n d^n$ grammars, showing that performance varied significantly with sequence length sampling strategies.

Beyond traditional RNNs, the role of specific activation functions in enhancing expressivity has also been studied. Weiss et al. (2018b) showed that RNNs with ReLU activations are strictly more powerful than those using standard sigmoid or tanh activations when it comes to counting tasks. This observation suggests that architectural modifications can

significantly alter the network's functional capacity. In a similar vein, Stogin et al. (2024) proposed neural network pushdown automata and neural network Turing machines, establishing a theoretical framework for integrating stacks into neural architectures, thereby enabling them to simulate complex computational models like pushdown automata and Turing machines. On the stability and generalization front, Dave et al. (2024) compared the stability of states learned by first-order and second-order RNNs when trained on Tomita and Dyck grammars. Their results indicate that second-order RNNs are better suited for maintaining stable state representations across different grammatical tasks, which is critical for ensuring that the learned model captures the true structure of the language. Their work also explored methods for extracting deterministic finite automata (DFA) from trained networks, evaluating the effectiveness of extraction techniques like those proposed by Weiss et al. (2018a) and Wang et al. (2018). This line of research is pivotal in understanding how well trained RNNs can be interpreted and how their internal state representations correspond to formal structures.

In terms of language generation and hierarchical structure learning, Hewitt et al. (2020) demonstrated that LSTMs, when trained as language generators, can learn Dyck-$(k, m)$ languages, which involve hierarchical and nested dependencies, drawing parallels between these formal languages and syntactic structures in natural languages. Finally, several studies have shown that the choice of objective functions and learning algorithms significantly affects RNNs' ability to stably learn complex grammars. For instance, Lan et al. (2022) and Mali et al. (2021) demonstrated that specialized loss functions, such as minimum description length, lead to more stable convergence and better generalization on formal language tasks. In light of the diverse findings from the aforementioned studies, our work systematically analyzes the divergence between the theoretical expressivity of RNNs and their empirical generalization capabilities through the lens of fixed-point theory. Specifically, we investigate how different RNN architectures capture and maintain stable state representations when learning complex grammars, focusing on the role of numerical precision, learning dynamics, and model stability. By leveraging theoretical results on fixed points and state stability, we provide a unified framework to evaluate the strengths and limitations of various RNN architectures.

## 3. Fixed Points of Discriminant Functions

In this section we focus on two prominent discriminant functions: *sigmoid* and *tanh*, both of which are extensively utilized in widely-adopted RNN cells such as LSTM and O2RNN.

**Theorem 1** *BROUWER'S FIXED POINT THEOREM* Boothby (1971): *For any continuous mapping $f : Z \to Z$, where $Z$ is a compact, non-empty convex set, $\exists\, z_f$ s.t. $f(z_f) \to z_f$*

From *Brouwer's fixed point theorem*, we directly get the following corollaries:

**Corollary 2** *Let $f : \mathbb{R} \to \mathbb{R}$ be a continuous, monotonic function with a non-empty, bounded, and convex co-domain $\mathbb{D} \subset \mathbb{R}$. Then $f$ has at least one fixed point, i.e., there exists some $c \in \mathbb{R}$ such that $f(c) = c$.*

**Corollary 3** *A parameterized sigmoid function of the form $\sigma(x) = \frac{1}{1+e^{-(wx+b)}}$, where $w, b \in \mathbb{R}$, has at least one fixed point, i.e., there exists some $c \in \mathbb{R}$ such that $\sigma(c) = c$.*

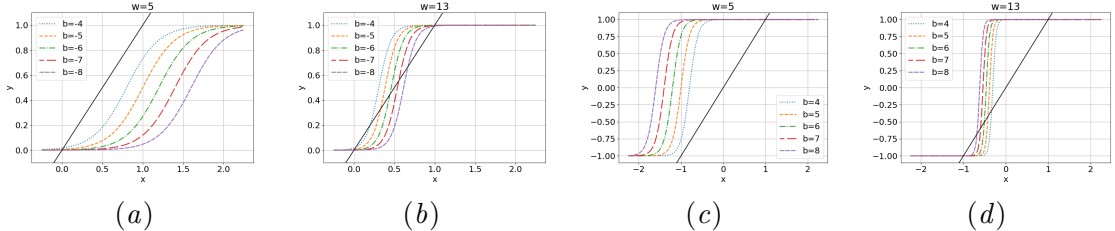

$$(a) \qquad (b) \qquad (c) \qquad (d)$$

Figure 1: The fixed points of discriminant function $f(wx + b)$ are the intersection points with the line $g(x) = x$ (solid black curve). The given figures show the existence of fixed points for $b$ in range $[-8, -4]$ and $w = 13$ for *sigmoid* ( a and b) and *tanh* ( c and d ). We can observe here that in the given range as the $w$ increased from 5 to 13, the number of fixed points increased from 1 to 3.

**Corollary 4** *A parameterized* tanh *function of the form* $\gamma(x) = \tanh(wx+b)$, *where* $w, b \in \mathbb{R}$, *has at least one fixed point, i.e., there exists some* $c \in \mathbb{R}$ *such that* $\gamma(c) = c$.

In this work we go beyond sigmoid and show that tanh also has three fixed points. Figure 1 visually demonstrates the existence of three fixed points for sigmoid as well as tanh discriminant functions.

**Theorem 5** *A parameterized* tanh *function* $\gamma(x) = \tanh(wx + b)$ *has three fixed points for a given* $b \in ]b^-, b^+[$ *and* $w > w_b$ *for some* $b^-, b^+, w_b \in \mathbb{R}$ *and* $b^- < b^+$.

The proofs for the theorem and the above corollaries are provided in the Appendix.

## 4. Experiment Setup

### Models

We evaluate the performance of two types of Recurrent Neural Networks (RNNs): Long Short-Term Memory (LSTM) networks by Hochreiter and Schmidhuber (1997) and Second-Order Recurrent Neural Networks (O2RNNs) by Omlin and Giles (1992a). The LSTM is considered a first-order RNN since its weight tensors are second-order matrices, whereas the O2RNN utilizes third-order weight tensors for state transitions, making it a second-order RNN. The state update for the O2RNN is defined as follows:

$$h_i^{(t)} = \sum_{j,k} w_{ijk} x_j^{(t)} h_k^{(t-1)} + b_i,$$

where $w_{ijk}$ is a third-order tensor that models the interactions between the input vector $x^{(t)}$ and the previous hidden state $h^{(t-1)}$, and $b_i$ is the bias term. All models consist of a single recurrent layer followed by a sigmoid activation layer for binary classification, as defined in Equation 10.

**Datasets**

We conduct experiments on eight different formal languages, divided into two categories: Dyck languages and counter languages. The Dyck languages include Dyck-1, Dyck-2, Dyck-4, and Dyck-6, which vary in the complexity and depth of nested dependencies. The counter languages include $a^n b^n c^n$, $a^n b^n c^n d^n$, $a^n b^m a^m b^n$, and $a^n b^m a^m b^m$. Each language requires the network to learn specific counting or hierarchical patterns, posing unique challenges for generalization.

The number of neurons used in the hidden state for each RNN configuration is summarized in Table 1. To ensure robustness and a fair comparison, all models were trained on sequences with lengths ranging from 1 to 40 and tested on sequences of lengths ranging from 41 to 500, thereby evaluating their generalization capability on longer and more complex sequences.

**Training and Testing Methodology**

Since the number of possible sequences grows exponentially with length (for a sequence of length $l$, there are $2^l$ possible combinations), we sampled sequences using an inverse exponential distribution over length, ensuring a balanced representation of short and long strings during training. Each model was trained to predict whether a given sequence is a positive example (belongs to the target language) or a negative example (does not follow the grammatical rules of the language).

For all eight languages, positive examples are inherently sparse in the overall sample space. This sparsity makes the generation of negative samples crucial to ensure a challenging and informative training set. We generated three different datasets, each using a distinct strategy for sampling negative examples:

1. ***Hard 0 (Random Sampling)***: Negative samples were randomly generated from the sample space without any structural similarity to positive samples. This method creates a broad variety of negatives, but many of these are trivially distinguishable, providing limited learning value for more sophisticated models.

2. ***Hard 1 (Edit Distance Sampling)***: Negative samples were constructed based on their string edit distance from positive examples. Specifically, for a sequence of length $l$, we generated negative strings that have a maximum edit distance of $0.25l$. This approach ensures that negative samples are structurally similar to positive ones, making it challenging for the model to differentiate them based solely on surface-level patterns.

3. ***Hard 2 (Topological Proximity Sampling)***: Negative samples were generated using topological proximity to positive strings, based on the structural rules of the language. For instance, in the counter language $a^n b^n c^n$, a potential negative string could be $a^{n-1} b^{n+1} c^n$, which maintains a similar overall structure but violates the language's grammatical constraints. This method ensures that the negative samples are more nuanced, requiring the model to maintain precise state transitions and counters to correctly classify them.

## 5. Results and Discussion

| model → | | lstm | | lstm | | o2rnn | | o2rnn | |
| **(trained layers)** | | (all layers) | | (classifier-only) | | (all layers) | | (classifier-only) | |
| **grammars** | **sdim** | **max** | **mean±std** | **max** | **mean±std** | **max** | **mean±std** | **max** | **mean±std** |
|---|---|---|---|---|---|---|---|---|---|
| Dyck-1 | 2 | 85.95 | $82.88 \pm 2.54$ | 73.89 | $72.3 \pm 1.28$ | 83.38 | $80.57 \pm 2.37$ | 63.88 | $63.85 \pm 0.02$ |
| Dyck-2 | 4 | 98.65 | $87.35 \pm 8.3$ | 72.99 | $70.05 \pm 3.35$ | 86.85 | $82.57 \pm 2.11$ | 63.67 | $60.75 \pm 2.41$ |
| Dyck-4 | 8 | 99.2 | $86.57 \pm 8.01$ | 71.93 | $69.58 \pm 3.71$ | 99.11 | $94.55 \pm 0.85$ | 63.71 | $59.88 \pm 2.44$ |
| Dyck-6 | 12 | 97.45 | $87.34 \pm 8.85$ | 72.27 | $68.64 \pm 2.24$ | 99.54 | $98.4 \pm 1.42$ | 62.74 | $60.33 \pm 1.79$ |
| $a^n b^n c^n$ | 6 | 98.13 | $90.17 \pm 15.30$ | 81.09 | $78.60 \pm 1.54$ | 97.86 | $97.27 \pm 0.35$ | 69.71 | $69.66 \pm 0.03$ |
| $a^n b^n c^n d^n$ | 8 | 98.33 | $90.45 \pm 14.22$ | 80.95 | $79.59 \pm 0.97$ | 97.24 | $96.11 \pm 2.43$ | 71.8 | $71.74 \pm 0.03$ |
| $a^n b^m a^m b^n$ | 8 | 99.83 | $98.05 \pm 4.65$ | 70.83 | $69.69 \pm 0.74$ | 99.76 | $99.58 \pm 0.19$ | 58.7 | $58.08 \pm 0.48$ |
| $a^n b^m a^m b^m$ | 8 | 99.93 | $99.64 \pm 0.56$ | 73.25 | $70.59 \pm 1.18$ | 99.43 | $99.13 \pm 0.38$ | 58.67 | $56.95 \pm 2.58$ |

Table 1: Performance comparison of RNNs trained with all layers and when trained with all weights frozen except classifier with *hard 1* negative string sampling.

| | | lstm | | lstm | | o2rnn | | o2rnn | |
| | | (all layers) | | (classifier-only) | | (all layers) | | (classifier-only) | |
| **gram.** | **n. set** | **max** | **mean ± std** | **max** | **mean ± std** | **max** | **mean ± std** | **max** | **mean ± std** |
|---|---|---|---|---|---|---|---|---|---|
| | hard 0 | 99.92 | $99.28 \pm 0.29$ | 96.13 | $93.14 \pm 2.53$ | 99.61 | $99.27 \pm 0.48$ | 83.37 | $83.32 \pm 0.03$ |
| $a^n b^n c^n$ | hard 1 | 98.13 | $90.17 \pm 15.30$ | 81.09 | $78.60 \pm 1.54$ | 97.86 | $97.27 \pm 0.35$ | 69.71 | $69.66 \pm 0.03$ |
| | hard 2 | 87.49 | $74.35 \pm 13.23$ | 75.64 | $74.48 \pm 0.73$ | 86.42 | $82.10 \pm 3.3$ | 69.94 | $69.85 \pm 0.07$ |
| | hard 0 | 99.59 | $99.36 \pm 0.19$ | 98.1 | $95.94 \pm 1.49$ | 99.48 | $98.91 \pm 1.17$ | 87.53 | $87.5 \pm 0.02$ |
| $a^n b^n c^n d^n$ | hard 1 | 98.33 | $90.45 \pm 14.22$ | 80.95 | $79.59 \pm 0.97$ | 97.24 | $96.11 \pm 2.43$ | 71.8 | $71.74 \pm 0.03$ |
| | hard 2 | 85.81 | $71.66 \pm 12.21$ | 75.33 | $74.47 \pm 1.29$ | 85.61 | $80.84 \pm 3.68$ | 70.72 | $70.66 \pm 0.03$ |

Table 2: Performance of RNNs declines when negative strings closer to positive strings are sampled for training

### Learnability of Dyck and Counter Languages

The results from Table 2 for negative set *hard 0* confirm prior findings on the expressivity of LSTMs and RNNs on counter, context-free, and context-sensitive languages. A one-layer LSTM is theoretically capable of representing all classes of counter languages, indicating that its expressivity is sufficient to model non-regular grammars. However, the results for negative sets *hard 1* and *hard 2* indicate that this expressivity does not necessarily translate to practical learnability. The observed performance drop on these harder negative sets suggests that, despite the LSTM's capacity to model such languages, its ability to generalize correctly under realistic training conditions is limited. This discrepancy between expressivity and learnability calls for a deeper understanding of how the network's internal dynamics align with the objective function during training.

In particular, the sparsity of positive samples combined with naively sampled negative examples (as in *hard 0*) allows the classifier to partition the feature space even when the internal feature encodings are not well-structured. This may give an inflated impression of the LSTM's practical learnability. Tables 1 and 2 compare fully trained models and classifier-only trained models, showing that the latter can achieve above-chance accuracy, even with minimal feature encoding. When negative samples are sampled closer to positive ones, as in *hard 1* and *hard 2*, the classifier struggles to maintain robust partitions, highlighting that the underlying feature encodings are not sufficiently aligned with the grammar

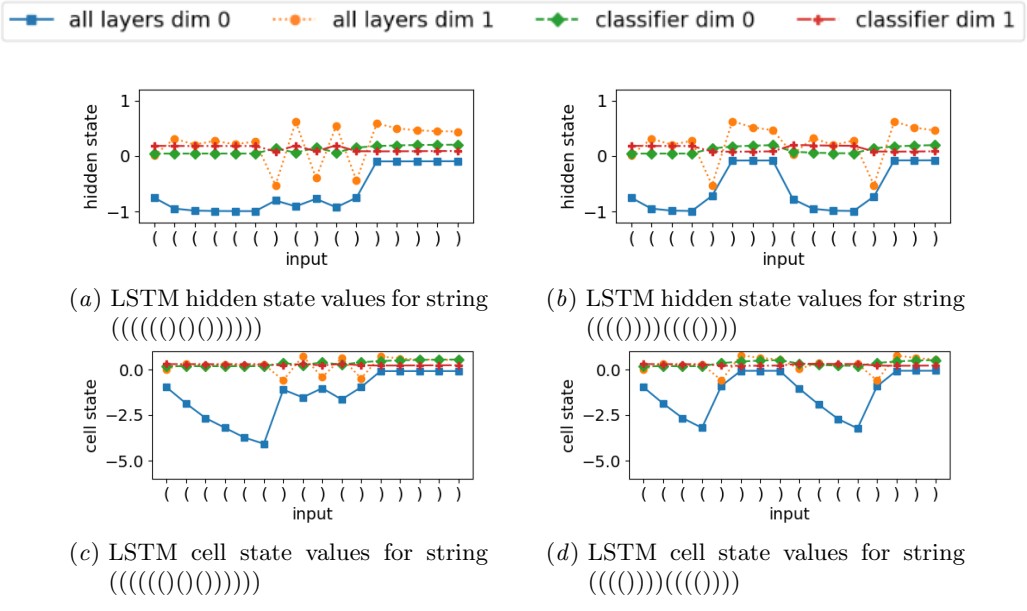

(a) LSTM hidden state values for string
    ((((((()()())))))))

(b) LSTM hidden state values for string
    (((()))) (((())))

(c) LSTM cell state values for string
    ((((((()()())))))))

(d) LSTM cell state values for string
    (((()))) (((())))

Figure 2: Transitions of hidden and cell states of LSTM for Dyck-1 grammars

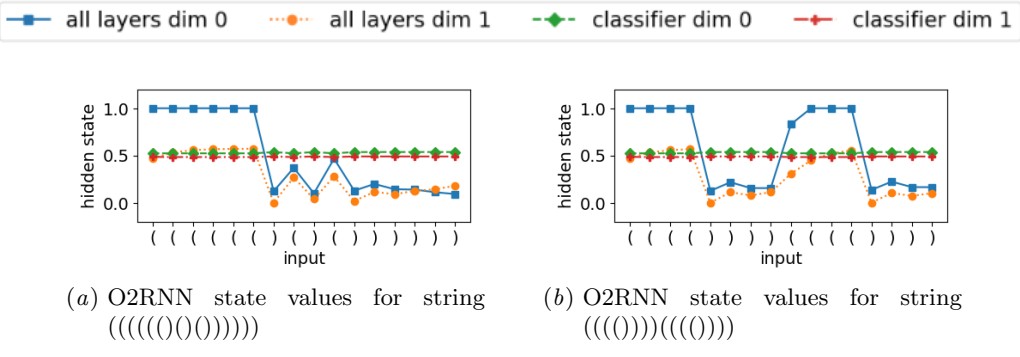

(a) O2RNN state values for string
    ((((((()()())))))))

(b) O2RNN state values for string
    (((()))) (((())))

Figure 3: Transitions of hidden states of O2RNN for Dyck-1 grammars

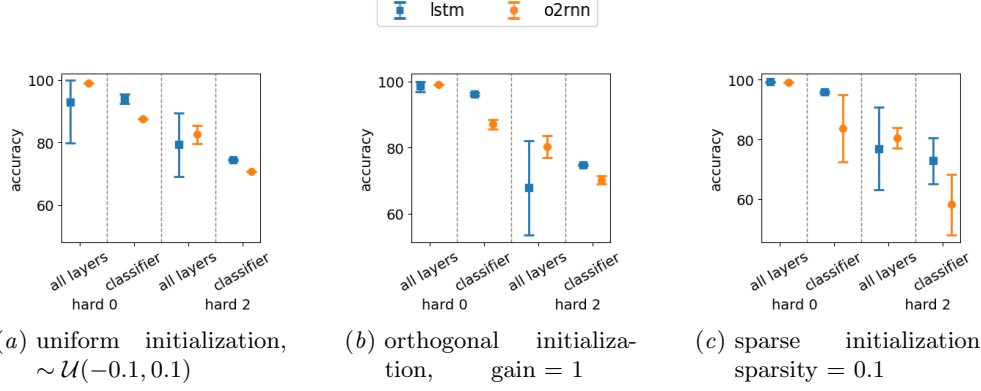

(a) uniform initialization,
    $\sim \mathcal{U}(-0.1, 0.1)$

(b) orthogonal initialization, gain $= 1$

(c) sparse initialization, sparsity $= 0.1$

Figure 4: Performance of LSTM and O2RNN with various weight initialization strategies on $a^n b^n c^n d^n$

structure. Future work can leverage fixed-point theory and expressivity analysis to establish better learnability bounds, offering a more principled approach to bridge the gap between theoretical capacity and empirical generalization.

## Stability of Feature Encoding in LSTM

The stability of LSTM feature encodings is heavily influenced by the precision of the network's internal dynamics. Across 10 random seeds, the standard deviation of accuracy for fully trained LSTMs is significantly higher compared to classifier-only models, particularly for challenging sampling strategies. For example, a fully trained LSTM on $a^n b^n c^n$ shows a standard deviation of 15.30% compared to only 1.54% for the classifier-only network using *hard 1* sampling. This difference is less pronounced for *hard 0* (0.29%) but becomes more severe for *hard 2* (13.23%), indicating that instability in the learned feature encodings increases as the negative examples become structurally closer to positive ones. This instability is due to the LSTM's reliance on its cell state to encode dynamic counters, which may not align precisely with the hidden state used for classification. As a result, slight deviations in internal dynamics cause substantial fluctuations in performance, suggesting a lack of robust fixed-point behavior in the cell state.

## Stability of Second-Order RNNs

In contrast, the O2RNN, which utilizes a third-order weight tensor, demonstrates more consistent performance across different training strategies and random seeds. In all configurations, the O2RNN exhibits a standard deviation of less than 4%, as shown in Tables 1 and 2. This stability is attributed to the higher-order interactions in the weight tensor, which drive the activation dynamics towards more stable fixed points. The convergence to these stable fixed points results in more robust internal state representations, making the O2RNN less sensitive to variations in training data and initialization. These findings are consistent with observations by Dave et al. (2024) for regular and Dyck languages, suggesting that higher-order tensor interactions inherently stabilize the internal dynamics, improving the alignment between the learned state transitions and the theoretical expressive capacity.

## Dynamic Counting and Fixed Points

The LSTM's ability to perform dynamic counting is closely tied to the stability of its cell state, which relies on the fixed points of the *tanh* activation function, as shown in Equation 11. Figures 2(*c*) and 2(*d*) provide evidence of dynamic counting when the LSTM encounters consecutive open brackets, as indicated by the solid blue curve that decreases monotonically. This behavior is in accordance with Equation 12, where the hidden state saturates to -1. However, when the network encounters a closing bracket, the cell state counter collapses, causing the hidden and cell states to start mirroring each other. This collapse occurs due to a mismatch between the counter dynamics and the LSTM's training objective, which primarily optimizes for hidden state changes rather than directly influencing the cell state's stability.

The root cause lies in the misalignment between the counter dynamics and the classification objective. Since the classification layer only uses the hidden state as input, any instability in the cell dynamics propagates through the hidden state, making it difficult

for the network to maintain precise counter updates. In contrast, the O2RNN's pure state approximation mechanism, as illustrated in Figure 3, shows smoother transitions and stable dynamics, indicating that the network's internal states are better aligned with its expressivity requirements.

### Effect of Initialization on Fixed-Point Stability

The choice of initialization strategy significantly influences the stability of fixed points in RNNs. Figure 4 shows that the performance of both LSTMs and O2RNNs declines from *hard 0* to *hard 2* for all initialization strategies. However, we observe that the O2RNN is particularly sensitive to sparse initialization, while being more stable for the other two initialization methods. This sensitivity reflects the network's reliance on precise weight configurations to drive its activation dynamics towards stable fixed points. In contrast, the LSTM's performance is relatively invariant to initialization strategies, as the collapse of its counting dynamics is more directly influenced by interactions between its gates rather than by initial weight values. Understanding the role of initialization in achieving stable fixed-point dynamics is crucial for designing networks that can consistently maintain dynamic behaviors throughout training.

## 6. Conclusion

Our framework analyzed models based on the fixed-point theory of activation functions and the precision of classification, providing a unified approach to study the stability and learnability of recurrent networks. By leveraging this framework, we identified critical gaps between the theoretical expressivity and the empirical learnability of LSTMs on Dyck and counter languages. While the LSTM cell state theoretically has the capacity to implement dynamic counting, we observed that misalignment between the training objective and the network's internal state dynamics often causes a collapse of the counter mechanism. This collapse leads the LSTM to lose its counting capacity, resulting in unstable feature encodings in its final state representations. Additionally, our analysis showed that this instability is masked in standard training setups due to the power of the classifier to partition the feature space effectively. However, when the dataset includes closely related positive and negative samples, this instability prevents the network from maintaining clear separations between similar classes, ultimately resulting in a decline in performance. These findings underscore that, despite LSTMs' theoretical capability for complex pattern recognition, their practical performance is hindered by internal instability and sensitivity to training configurations. To address this gap, our fixed-point analysis focused on understanding the stability of activation functions, offering a mathematical framework that connects theoretical properties to empirical behaviors. This approach provides new insights into how activation stability can influence the overall learnability of a system, enabling us to better align theory and practice. Our results emphasize that improving the stability of counter dynamics in LSTMs can lead to more robust, generalizable memory-augmented networks. Ultimately, this work contributes to a deeper understanding of the learnability of LSTMs and other recurrent networks, paving the way for future research that bridges the divide between theoretical expressivity and practical generalization.

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

## Appendix A. Appendix: Additional Results and Discussion

### A.1. Training Details

For reproducibility and stability we train each model over 10 seeds and report the mean, standard deviation, and maximum of accuracy over test set. We use stochastic gradient descent optimizer for a maximum of $100,000$ iterations. We employ a batch size of 128 and a learning rate of 0.01. Validation is run very 100 iterations and training is stopped if validation loss does not improve for 7000 consecutive iterations. All models use binary cross entropy loss as optimization function.

We use uniform random initialization ($\mathcal{U}(-\sqrt{k}, \sqrt{k})$, where $k$ is the hidden size) for LSTM weights and normal initialization ($\mathcal{N}(0, 0.1)$) for O2RNN for all experiments, except figure 4 which compares performance of LSTM and O2RNN on $a^n b^n c^n d^n$ with the following initialization strategies: 1) *uniform initialization* : $w \sim \mathcal{U}(-0.1, 0.1)$, 2) *orthogonal initialization* : gain $= 1$, and 3) *sparse initialization* : sparsity $= 0.1$, all non zero $w_{ij}$ are sampled from $\mathcal{N}(0, 0.01)$. All the biases are initialized with a constant value of 0.01

Tables 1 and 2 show results comparing models trained in two different ways : (1) *all layers*: all layers of the model are trained, and (2) *classifier-only*: weights of the RNN cells are frozen after random initialization and only classifier is trained.

We use Nvidia 2080ti GPUs to run our experiments with training times varying from under 15 minutes for a simpler dataset like Dyck-1 on O2RNN, to over 60 minutes for a counter language on LSTM. In total, we train 700 models for our main results with over 400 hours of cumulative GPU training times.

### A.2. Generalization Results

Figure 5 shows the generalization plots for LSTM and O2RNN for both training strategies i.e. all layers trained and classifier-only trained. These networks were trained on string lengths $2-40$ and tested on lengths $41-500$. The plots show the distribution of performance across the test sequence lengths. Both RNNs maintain their accuracy across the test range indicating generalization of the results.

### A.3. Results with Transformers

To examine the capacity of transformer encoder architecture and compare them with our results from RNNs, we train one layer transformer encoder architecture. For binary classification of counter languages, we adopt two different embedding strategies as input to the classifier:

1. *transformer-avg* : The classification layer receives the mean of all output embeddings generated by the transformer encoder as input feature.

2. *transformer-cls* : The classification layer receives the output embedding of [CLS] token as input feature.

We train single layer transformer encoder network on two counter languages $a^n b^n c^n$ and $a^n b^n c^n d^n$. We use the embedding dimension of 8 with 4 attention heads. Table 5 and figure 6 shows that one-layer transformer encoder model fails to learn counter languages.

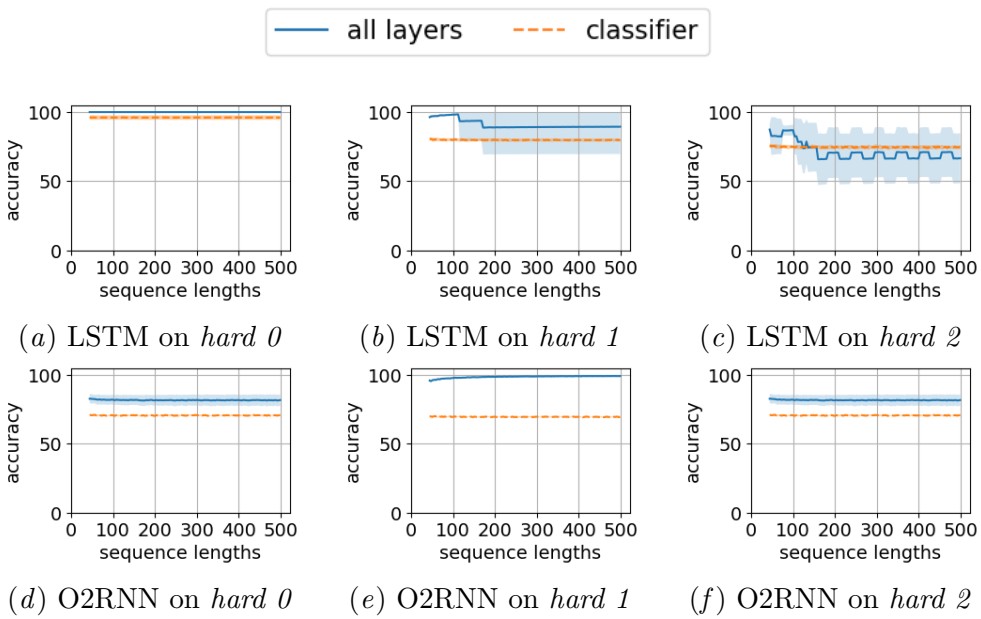

(a) LSTM on *hard 0*    (b) LSTM on *hard 1*    (c) LSTM on *hard 2*

(d) O2RNN on *hard 0*   (e) O2RNN on *hard 1*   (f) O2RNN on *hard 2*

Figure 5: Generalization plots for LSTM and O2RNN

Among the two classification strategies, *transformer-cls* shows high standard deviation in performance than *transformer-avg* across 10 seeds. transformer-cls model on some seed performed as high as 64% on $a^n b^n c^n d^n$ grammars, however the mean performance across 10 seeds remained near 50%. *transformer-cls* model does not show any signs of training (table 4) for weight initialization strategies used for comparing RNNs. For most seeds, the network has 50% accuracy.

### A.4. Results on Penn Tree Bank dataset

Table 3 compares O2RNN, LSTM and one-layer transformer encoder network on PTB dataset. O2RNN and LSTM are trained with hidden state size of 8 for character level training, and with size 256 for word level training. For transformer-encoder model we use similar embedding dimensions - 8 for character level training and 256 for word level training.

| dataset | model | all layers | classifier-only |
|---------|-------|------------|-----------------|
| | lstm | 3.1243 | 7.7886 |
| ptb-char | o2rnn | 3.2911 | 8.4865 |
| | transformer-
-encoder | 4.4389 | 9.6622 |
| | lstm | 160.3073 | 403.9483 |
| ptb-word | o2rnn | 283.5615 | 356.4486 |
| | transformer-
-encoder | 196.8097 | 318.425 |

Table 3: Perplexity of LSTM, O2RNN, and transformer-encoder models on PTB dataset with all layers trained and classifier-only training.

| initialization | negative samples | model | all layers | | classifier | |
|---|---|---|---|---|---|---|
| | | | max | mean ± std | max | mean ± std |
| uniform | hard 0 | lstm | 99.89 | 92.96 ± 13.2 | 95.54 | 94.03 ± 1.48 |
| | | o2rnn | 99.12 | 99.1 ± 0.01 | 87.54 | 87.5 ± 0.02 |
| | | transformer-cls | 50 | 50.00 ± 0.00 | 50 | 49.99 ± 0.02 |
| | hard 2 | lstm | 86.96 | 79.33 ± 10.15 | 74.74 | 74.41 ± 0.31 |
| | | o2rnn | 85.5 | 82.6 ± 2.93 | 70.72 | 70.66 ± 0.03 |
| | | transformer-cls | 50 | 50.00 ± 0.00 | 50 | 50.00 ± 0.00 |
| orthogonal | hard 0 | lstm | 99.99 | 98.64 ± 1.76 | 96.88 | 96.13 ± 0.5 |
| | | o2rnn | 99.12 | 99.10 ± 0.01 | 87.54 | 87.03 ± 1.41 |
| | | transformer-cls | 53.05 | 50.58 ± 1.0 | 50.22 | 49.99 ± 0.13 |
| | hard 2 | lstm | 86.55 | 67.85 ± 14.15 | 75.24 | 74.83 ± 0.27 |
| | | o2rnn | 85.41 | 80.34 ± 3.36 | 70.72 | 70.25 ± 1.23 |
| | | transformer-cls | 51.23 | 50.15 ± 0.48 | 50.08 | 49.72 ± 0.61 |
| sparse | hard 0 | lstm | 99.59 | 99.27 ± 0.16 | 96.55 | 95.85 ± 0.39 |
| | | o2rnn | 99.12 | 99.10 ± 0.01 | 87.54 | 83.75 ± 11.25 |
| | | transformer-cls | 50 | 50.00 ± 0.00 | 50 | 50.00 ± 0.00 |
| | hard 2 | lstm | 86.26 | 76.96 ± 13.73 | 76.39 | 72.87 ± 7.63 |
| | | o2rnn | 85.66 | 80.50 ± 3.48 | 70.72 | 58.26 ± 10.12 |
| | | transformer-cls | 50 | 50.00 ± 0.00 | 50 | 50.00 ± 0.00 |

Table 4: Effect of weight initialization strategies on networks ability to respond to topologically close positive and negative strings

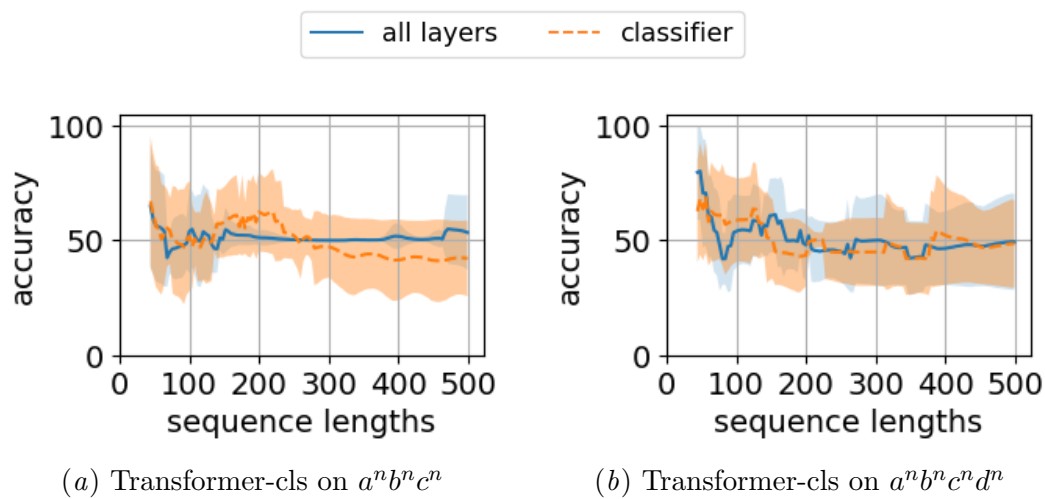

(a) Transformer-cls on $a^n b^n c^n$      (b) Transformer-cls on $a^n b^n c^n d^n$

Figure 6: Generalization plots for *transformer-cls* network with *hard 0* negative sampling strategy

| grammar | feature | layers trained | max | mean $\pm$ std |
|---------|---------|----------------|-----|----------------|
| $a^n b^n c^n$ | cls | all layers | 55.45 | $51.70 \pm 2.37$ |
| | | classifier-only | 60.22 | $49.88 \pm 7.40$ |
| | avg-pool | all layers | 58.58 | $51.53 \pm 2.44$ |
| | | classifier-only | 59.15 | $51.60 \pm 2.92$ |
| $a^n b^n c^n d^n$ | cls | all layers | 64.25 | $51.99 \pm 10.59$ |
| | | classifier-only | 60.47 | $50.22 \pm 9.38$ |
| | avg-pool | all layers | 55.79 | $49.35 \pm 3.01$ |
| | | classifier-only | 52.36 | $49.41 \pm 1.27$ |

Table 5: One layer transformer encoder networks do not learn counter languages like $a^n b^n c^n$ and $a^n b^n c^n d^n$ with *hard 0* negative sampling strategy

## Appendix B. Counter Machines

Counter machines Fischer et al. (1968) are abstract machines composed of finite state automata controlling one or more counters. A counter can either increment $(+1)$, decrement $(-1$ if $> 0)$, clear $(\times 0)$, do nothing $(+0)$. A counter machine can be formally defined as:

**Definition 6** *A counter machine (CM) is a 7-tuple* $(Q, \Sigma, q_0, F, \delta, \gamma, \mathbf{1}_{=0})$ *where*

- $\Sigma$ *is a finite alphabet*

- $Q$ *is a set of states with* $q_0 \in Q$ *as initial state*

- $F \subset Q$ *is a set of accepting states.*

- $\mathbf{1}_{=0}$ *checks the state of the counter and returns* $1$ *if counter is zero else returns* $0$

- $\gamma$ *is the counter update function defined as:*

$$\gamma : \Sigma \times Q \times \mathbf{1}_{=0} \to \{\times 0, -1, +0, +1\} \tag{1}$$

- $\delta$ *is the state transition function defined as:*

$$\delta : \Sigma \times Q \times \mathbf{1}_{=0} \to Q \tag{2}$$

Acceptance of a string in a counter machine can be assessed by either the final state is in $F$ or the counter reaches 0 at the end of the input.

## Appendix C. Learning to accept $a^n b^n$ with LSTM

LSTM Hochreiter and Schmidhuber (1997) is a gated RNN cell. The LSTM state is a tuple $(h, c)$ where $h$ is popularly known as hidden state and $c$ is known as cell state.

$$i_t = \sigma(W_i x_t + U_i h_{t-1} + b_i) \tag{3}$$

$$f_t = \sigma(W_f x_t + U_f h_{t-1} + b_f) \tag{4}$$

$$o_t = \sigma(W_o x_t + U_o h_{t-1} + b_o) \tag{5}$$

$$\tilde{c}_t = \tanh(W_c x_t + U_c h_{t-1} + b_c) \tag{6}$$

$$c_t = f_t \odot c_{t-1} + i_t \odot \tilde{c}_t \tag{7}$$

$$h_t = o_t \odot tanh(c_t) \tag{8}$$

A typical binary classification network with LSTM cell is composed of two parts:

1. The enocoder recurrent network

$$(h, c)_{t+1} = LSTM(x, (h, c)_t) \tag{9}$$

2. A classification layer, usually a single perceptron layer followed by a sigmoid

$$p = \sigma(W h_\tau + b) \tag{10}$$

In case of multiclass classification, $\sigma$ is replaced by *softmax* function.

Following the construction of Merrill et al. (2020) we can draw parallels between the workings of counter machine and LSTM cell. Here $i_t$ decides wheather to execute $+0$, while $\{+1, -1\}$ are decided by $\tilde{c}_t$. To execute $\times 0$ both $f_t$ and $i_t$ needs to be 0.

Also the cell state and hidden state of LSTM have *tanh* as discriminant function. In the case of $a^n b^n$, a continuous stream of $a$ is followed by an equal number of $b$, which creates an iterative execution of LSTM cell, making the output of discriminant functions closer to their fixed points. For maximum learnability, we can assume $\tilde{c}_t \in \{\xi^-, \xi^+\}$. Thus maximum final state values for $a^n b^n$ will be:

$$c_{a^n b^n} = n(\xi^+ + \xi^-) \tag{11}$$
$$h_{a^n b^n} = \tanh(n(\xi^+ + \xi^-)) \tag{12}$$

From the above equations we can see that, while cell state is unbounded, hidden state is bounded in range $]-1, 1[$. In our experiments we see that hidden state saturates to boundary values faster than is required to maintain the count. Formally, for some fairly moderate $\alpha$ and $\beta$ we can reach a point where $|\tanh(\alpha\xi^+ + \beta\xi^-) - \tanh(\alpha\xi^+ + (\beta+1)\xi^-)| < \epsilon$, where $\epsilon$ is the precision of the classification layer.

Saturation of hidden state is desired from the perspective of consistent calculation of counter updates. In the LSTM cell, saturated hidden state means more stable gates which in turn leads to consistent cell state. However from the perspective of the classification layer, a saturated hidden state does not offer much information for a robust classification.

**Expressivity of RNNs and DFA Equivalence:**

The expressivity of RNNs and even o2RNN is equivalent to that of deterministic finite automata (DFA) Merrill et al. (2020); Mali et al. (2023). In this context, the RNN's behavior mirrors that of a DFA, with distinct stable fixed points representing states for each input symbol. The transitions between these states are governed by the input sequence and the corresponding hidden state dynamics, which collapse to stable fixed points. This allows the RNN to encode complex grammars like $a^n b^n c^n$ purely through its internal state dynamics.

Thus it can be seen that RNN can encode the sequence $a^n b^n c^n$ by relying on the convergence of state dynamics to stable fixed points. The bounded sequence length $N$ ensures that the hidden states have sufficient time to converge to these fixed points, enabling the network to express such grammars within its capacity. The expressivity of the RNN, akin to a DFA, underlines that the encoding is achieved purely through state dynamics, which is especially true for 02RNN.

## Appendix D. Fixed Points of Discriminant Functions

In this section, we focus on two prominent discriminant functions: *sigmoid* and *tanh*, both of which are extensively utilized in widely adopted RNN cells such as LSTM and O2RNN.

**Theorem 7** *BROUWER'S FIXED POINT THEOREM Boothby (1971): For any continuous mapping $f : Z \to Z$, where $Z$ is a compact, nonempty convex set, $\exists z_f$ st. $f(z_f) \to z_f$*

**Corollary 8** *Let $f : \mathbb{R} \to \mathbb{R}$ be a continuous, monotonic function with a non-empty, bounded, and convex co-domain $\mathbb{D} \subset \mathbb{R}$. Then $f$ has at least one fixed point, that is, there exists some $c \in \mathbb{R}$ such that $f(c) = c$.*

**Proof**   Since $\mathbb{D} \subset \mathbb{R}$ is a non-empty, bounded, and convex set, let the codomain of $f$ be denoted as $\mathbb{D} = [a, b]$ for some $a, b \in \mathbb{R}$ with $a < b$. Consider the identity function $g(x) = x$, which is continuous on $\mathbb{R}$. The fixed points of $f$ correspond to the intersection points between $f(x)$ and $g(x)$, that is, the solutions to equation $f(x) = g(x)$.

Next, observe the behavior of $f$ outside its co-domain $\mathbb{D} = [a, b]$:

- For any $x < a$, we have $f(x) \geq a > x$ (since $f$ is monotonic), implying that $f(x) > x$.

- For any $x > b$, we have $f(x) \leq b < x$, implying that $f(x) < x$.

By the Intermediate Value Theorem, if $f(x) > x$ for some $x < a$ and $f(x) < x$ for some $x > b$, then there must exist a point $c \in [a, b]$ such that $f(c) = c$.

Thus, the function $f$ has at least one fixed point in the interval $[a, b]$.

∎

**Corollary 9** *A parameterized sigmoid function of the form $\sigma(x) = \frac{1}{1+e^{-(wx+b)}}$, where $w, b \in \mathbb{R}$, has at least one fixed point, i.e., there exists some $c \in \mathbb{R}$ such that $\sigma(c) = c$.*

**Proof**   Consider the function $\sigma(x) = \frac{1}{1+e^{-(wx+b)}}$. We want to show that $\sigma(x)$ has at least one fixed point. A fixed point is a value $c \in \mathbb{R}$ such that $\sigma(c) = c$.

First, observe that the sigmoid function $\sigma(x)$ is continuous and increases strictly for all $x \in \mathbb{R}$. The codomain of $\sigma(x)$ is the interval $[0, 1]$, i.e., $\sigma(x) \in [0, 1]$ for all $x \in \mathbb{R}$. We now consider the continuous identity function $g(x) = x$, which intersects the line $y = x$.

Next, let us analyze the behavior of $\sigma(x) - x$ as $x \to -\infty$ and $x \to +\infty$:

- As $x \to -\infty$, we have $e^{-(wx+b)} \to \infty$, which implies $\sigma(x) \to 0$. Therefore, $\sigma(x) - x \to -\infty$ as $x \to -\infty$.

- As $x \to +\infty$, we have $e^{-(wx+b)} \to 0$, which implies $\sigma(x) \to 1$. Therefore, $\sigma(x) - x \to 1 - x \to -\infty$ as $x \to +\infty$.

Since $\sigma(x) - x$ is a continuous function on $\mathbb{R}$ and changes sign (from positive to negative) as $x$ varies from $-\infty$ to $+\infty$, by the Intermediate Value Theorem, there must exist some $c \in \mathbb{R}$ such that:

$$\sigma(c) - c = 0 \quad \Rightarrow \quad \sigma(c) = c.$$

Hence, $\sigma(x)$ has at least one fixed point.

∎

**Corollary 10** *A parameterized* tanh *function of the form* $\gamma(x) = \tanh(wx + b)$, *where* $w, b \in \mathbb{R}$, *has at least one fixed point, i.e., there exists some* $c \in \mathbb{R}$ *such that* $\gamma(c) = c$.

**Proof** Consider the function $\gamma(x) = \tanh(wx + b)$. We want to show that $\gamma(x)$ has at least one fixed point. A fixed point is a value $c \in \mathbb{R}$ such that $\gamma(c) = c$.

Step 1: Properties of the Function $\gamma(x)$ The hyperbolic tangent function, $\tanh(x)$, is a continuous and strictly increasing function for all $x \in \mathbb{R}$. For any real value $y$, the function $\tanh(y)$ is bounded and satisfies $-1 < \tanh(y) < 1$. Thus, the co-domain of $\gamma(x) = \tanh(wx + b)$ is also bounded within $[-1, 1]$, i.e., $\gamma(x) \in [-1, 1]$ for all $x \in \mathbb{R}$.

Furthermore, since $\tanh(x)$ is strictly increasing, the function $\gamma(x) = \tanh(wx + b)$ is also strictly increasing in $x$. This implies that $\gamma(x)$ is one-to-one and continuous over $\mathbb{R}$.

Step 2: Analysis of $\gamma(x) - x$ Consider the function:

$$f(x) = \gamma(x) - x = \tanh(wx + b) - x.$$

We want to show that $f(x) = 0$ has at least one solution, i.e., there exists some $c \in \mathbb{R}$ such that $\tanh(wc+b) = c$. To analyze the existence of such a $c$, let us examine the behavior of $f(x)$ as $x \to \pm\infty$:

- As $x \to -\infty$: We have $wx + b \to -\infty$. Thus, $\tanh(wx + b) \to -1$. Therefore:

$$f(x) = \tanh(wx + b) - x \to -1 - x \to \infty.$$

- As $x \to +\infty$: We have $wx + b \to +\infty$. Thus, $\tanh(wx + b) \to 1$. Therefore:

$$f(x) = \tanh(wx + b) - x \to 1 - x \to -\infty.$$

Since $f(x)$ is continuous on $\mathbb{R}$ and changes sign from positive (as $x \to -\infty$) to negative (as $x \to +\infty$), by the Intermediate Value Theorem, there must exist some $c \in \mathbb{R}$ such that:

$$f(c) = \tanh(wc + b) - c = 0.$$

This implies that $\gamma(c) = c$, i.e., $\gamma(x)$ has at least one fixed point.

∎

Prior Omlin and Giles (1996) have shown that parameterized sigmoid function $\sigma(x) = \frac{1}{1+e^{-(wx+b)}}$ has three fixed points for a given $b \in ]b^-, b^+[$ and $w > w_b$ for some $b^-, b^+, w_b \in \mathbb{R}$ and $b^- < b^+$. Further they showed sigmoid has two stable fixed point. In this work we go beyond sigmoid and show that TanH also has three fixed points

**Theorem 11** *A parameterized* tanh *function* $\gamma(x) = \tanh(wx + b)$ *has three fixed points for a given* $b \in ]b^-, b^+[$ *and* $w > w_b$ *for some* $b^-, b^+, w_b \in \mathbb{R}$ *and* $b^- < b^+$.

**Proof** We start by defining a fixed point of the function $\gamma(x) = \tanh(wx + b)$. A fixed point $x$ satisfies the equation:

$$\gamma(x) = x \quad \Rightarrow \quad \tanh(wx + b) = x.$$

Let us define a new function to analyze the fixed points:

$$f(x) = \tanh(wx + b) - x.$$

The fixed points of $\gamma(x)$ are the solutions to the equation $f(x) = 0$. We will analyze $f(x)$ in detail to determine the number of solutions.

**Step 1: Properties of $f(x)$**

The function $f(x) = \tanh(wx + b) - x$ is continuous and differentiable. We start by computing its derivative:

$$f'(x) = \frac{d}{dx}\left(\tanh(wx + b) - x\right) = w \cdot \operatorname{sech}^2(wx + b) - 1,$$

where $\operatorname{sech}(y) = \frac{2}{e^y + e^{-y}}$ is the hyperbolic secant function. The value of $\operatorname{sech}^2(wx + b)$ satisfies $0 < \operatorname{sech}^2(y) \leq 1$. Thus:

$$f'(x) = w \cdot \operatorname{sech}^2(wx + b) - 1.$$

**Step 2: Critical Points of $f(x)$**

The critical points occur when $f'(x) = 0$:

$$w \cdot \operatorname{sech}^2(wx + b) - 1 = 0 \quad \Rightarrow \quad \operatorname{sech}^2(wx + b) = \frac{1}{w}.$$

Since $0 < \operatorname{sech}^2(wx + b) \leq 1$, the above equation has a real solution if and only if:

$$w > 1.$$

For $w > 1$, there are exactly two critical points, $x_1$ and $x_2$, such that $x_1 < x_2$.

**Step 3: Behavior of $f(x)$ as $x \to \pm\infty$**

As $x \to \infty$, $wx + b \to \infty$ for $w > 0$. Thus, $\tanh(wx + b) \to 1$. Hence:

$$f(x) = \tanh(wx + b) - x \to 1 - x \quad \text{as} \quad x \to \infty.$$

Therefore, $f(x) \to -\infty$ as $x \to \infty$.

As $x \to -\infty$, $wx + b \to -\infty$ for $w > 0$. Thus, $\tanh(wx + b) \to -1$. Hence:

$$f(x) = \tanh(wx + b) - x \to -1 - x \quad \text{as} \quad x \to -\infty.$$

Therefore, $f(x) \to \infty$ as $x \to -\infty$.

**Step 4: Intermediate Value Theorem**

The intermediate value theorem tells us that since $f(x)$ is continuous and changes sign from $\infty$ to $-\infty$, it must have at least one root. Thus, there is at least one fixed point for $\gamma(x)$.

**Step 5: Conditions for Three Fixed Points**

We want to show that for specific values of $b$ and $w$, the function $f(x) = \tanh(wx + b) - x$ has exactly three roots. To do so, we analyze $f(x)$ in detail around its critical points.

1. **Critical Points Analysis**:

Recall that the critical points of $f(x)$ are given by:

$$w \cdot \text{sech}^2(wx + b) - 1 = 0 \quad \Rightarrow \quad \text{sech}^2(wx + b) = \frac{1}{w}.$$

Let $y = wx + b$. Then the critical points $y_1$ and $y_2$ satisfy:

$$\text{sech}^2(y_1) = \frac{1}{w}, \quad \text{sech}^2(y_2) = \frac{1}{w}.$$

Solving for $y$, we get:

$$y_1 = \pm \cosh^{-1}(\sqrt{w}), \quad y_2 = -y_1.$$

Converting back to $x$:

$$x_1 = \frac{y_1 - b}{w}, \quad x_2 = \frac{y_2 - b}{w}.$$

2. **Local Minima and Maxima Analysis**:

At these critical points, the second derivative $f''(x)$ determines whether $f(x)$ has a local minimum or maximum:

$$f''(x) = w^2 \cdot (-2\text{sech}^2(wx + b)\tanh(wx + b)) - 1.$$

Analyzing $f''(x_1)$ and $f''(x_2)$, we can show that $x_1$ corresponds to a local minimum and $x_2$ corresponds to a local maximum (or vice-versa depending on $b$).

3. **Behavior of $f(x)$ in the Range $]b^-, b^+[$**:

For $b \in ]b^-, b^+[$ and $w > w_b$, $f(x)$ changes sign three times, indicating three distinct zeros.

Thus, for $w > w_b$ and $b \in ]b^-, b^+[$, the function $\gamma(x) = \tanh(wx + b)$ has exactly three fixed points.

■

This can be visualized in Figure 1(c, d). Let $b \in [-8, -4]$, then we can observe that $\gamma(x)$ meets $g(x) = x$ three times for $w = 13$, while it only meets $g(x) = x$ once for $w = 5$. Since $\gamma^{-1}(x)$ is also a monotonic function, and $w$ and $x$ will have a monotonic inverse relationship, for all $w \geq 13$, $\gamma(x)$ has 3 fixed points.

Next we show Tanh has out of three fixed point, two stable fixed points stable

**Theorem 12** *If a parameterized* tanh *function* $\gamma(x) = \tanh(wx + b)$ *has three fixed points* $\xi^-, \xi^0, \xi^+$ *such that* $-1 < \xi^- < \xi^0 < \xi^+ < 1$, *then* $\xi^-$ *and* $\xi^+$ *are stable fixed points.*

**Proof** Let us start by defining a fixed point of the function $\gamma(x) = \tanh(wx + b)$. A point $x$ is a fixed point if:

$$\gamma(x) = x \quad \Rightarrow \quad \tanh(wx + b) = x.$$

We are given that there are three fixed points $\xi^-, \xi^0, \xi^+$ such that:

$$-1 < \xi^- < \xi^0 < \xi^+ < 1.$$

Step 1: Stability Criterion for Fixed Points A fixed point $\xi$ is considered **stable** if the magnitude of the derivative of $\gamma(x)$ at $\xi$ is less than 1, i.e.,

$$|\gamma'(\xi)| < 1.$$

Conversely, a fixed point is **unstable** if:

$$|\gamma'(\xi)| > 1.$$

Step 2: Derivative of the Function $\gamma(x)$ We compute the derivative of $\gamma(x) = \tanh(wx + b)$:

$$\gamma'(x) = \frac{d}{dx}\left(\tanh(wx + b)\right).$$

Recall that the derivative of the hyperbolic tangent function is:

$$\frac{d}{dx}\tanh(x) = \operatorname{sech}^2(x),$$

where $\operatorname{sech}(x) = \frac{2}{e^x + e^{-x}}$. Using the chain rule, we obtain:

$$\gamma'(x) = w \cdot \operatorname{sech}^2(wx + b).$$

Thus, at a fixed point $\xi$, the derivative is:

$$\gamma'(\xi) = w \cdot \operatorname{sech}^2(w\xi + b).$$

Step 3: Stability Analysis at Each Fixed Point We will now analyze the derivative at each of the three fixed points to determine their stability.

1. Middle Fixed Point $\xi^0$:

Since $\xi^0$ is the middle fixed point, the function $\gamma(x)$ has a steep slope at $\xi^0$. Intuitively, the slope of $\tanh(x)$ around the origin (and for values near zero) is steep, making $|\gamma'(\xi^0)| > 1$. Thus, $\xi^0$ is an **unstable** fixed point.

2. Leftmost Fixed Point $\xi^-$:

Consider the derivative at the leftmost fixed point $\xi^-$:

$$\gamma'(\xi^-) = w \cdot \operatorname{sech}^2(w\xi^- + b).$$

Since $\xi^-$ is smaller in magnitude compared to $\xi^0$, the value of $\operatorname{sech}^2(w\xi^- + b)$ is close to 1 but slightly less, and thus:

$$|\gamma'(\xi^-)| < 1.$$

This implies that the fixed point $\xi^-$ is **stable**.

3. Rightmost Fixed Point $\xi^+$:

Similarly, for the rightmost fixed point $\xi^+$:

$$\gamma'(\xi^+) = w \cdot \text{sech}^2(w\xi^+ + b).$$

Since $\xi^+$ is greater than $\xi^0$, the value of $\text{sech}^2(w\xi^+ + b)$ is also close to 1 but less than at $\xi^0$, leading to:

$$|\gamma'(\xi^+)| < 1.$$

This means that the fixed point $\xi^+$ is **stable**.

Thus we have shown that for the three fixed points $\xi^-$, $\xi^0$, and $\xi^+$ of the function $\gamma(x) = \tanh(wx + b)$:

- $\xi^0$ is an **unstable** fixed point because $|\gamma'(\xi^0)| > 1$. - $\xi^-$ and $\xi^+$ are **stable** fixed points because $|\gamma'(\xi^-)| < 1$ and $|\gamma'(\xi^+)| < 1$.

Thus, the theorem is proven. ∎

We can make the following observations about the fixed points of *sigmoid* and *tanh* functions:

- If one fixed point exists, then it is a stable fixed point

- If two fixed point exists, then one fixed point is stable and other is unstable.

- If three fixed point exists, then two fixed points are stable and one is unstable.

### D.1. Stable and Unstable Fixed Points

$\sigma(wx + b)$ and $\tanh(wx + b)$ are monotonic functions with bounded co-domain. For $w > 0$, both functions are non-decreasing. Let $f : \mathbb{R} \to \mathbb{R}$ be a monotonic, non-decreasing function with bounded co-domain, and $g(x) = x$, $x \in \mathbb{R}$, Then,

- *If one fixed point exists, then it is a stable fixed point*
  Let $x > z_f$ where $z_f$ is the only fixed point. Then, $f(x) < g(x)$, thus iteratively $x_{i+1} = f(x_i)$, with each $x_{i+1} \leq x_i$ and equality occuring at $x_i = z_f$. Similary for $x < z_f$, we can show that with each iterative application of $f(x)$, $x$ moves towards $z_f$.

- *If two fixed points exists, then one fixed point is stable and other is unstable.*
  If there are two fixed points then at one fixed point $(z_t)$ $g(x)$ is tangent to $f(x)$. For $x \neq z_t$ $f(x) < g(x)$, thus making that fixed point unstable.

- *If three fixed point exists, then two fixed points are stable and one is unstable.*
  This is already shown in Theorem 3.2 and 3.3.

## Appendix E. Precision of Neural Network

Numerical precision plays an important role in the partition of feature space by the classifier network, especially when the final hidden state from RNN either collapses towards 0 or saturates asymptotically to the boundary values.

**Theorem 13** *Given a neural network layer with an input vector $\mathbf{H} \in \mathbb{R}^n$, a weight matrix $\mathbf{W} \in \mathbb{R}^{n \times m}$, a bias vector $\mathbf{b} \in \mathbb{R}^m$, and a sigmoid activation function $\sigma$, the output of the layer is defined by $f(\mathbf{H}) = \sigma(\mathbf{H} \cdot \mathbf{W} + \mathbf{b})$. The capacity of this layer to encode information is influenced by both the precision of the floating-point representation and the dynamical properties of the sigmoid function.*

*Let $\epsilon$ be the machine epsilon, which represents the difference between 1 and the least value greater than 1 that is representable in the floating-point system used by the network. Assume the elements of $\mathbf{W}$ and $\mathbf{b}$ are drawn from a Gaussian distribution and are fixed post-initialization.*

*Then, the following bounds hold for the output of the network layer:*

1. *The granularity of the output is limited by $\epsilon$, such that for any element $h_i$ in $\mathbf{H}$ and corresponding weight $w_{ij}$ in $\mathbf{W}$, the difference in the layer's output due to a change in $h_i$ or $w_{ij}$ less than $\epsilon$ may be imperceptible.*

2. *For $z = \mathbf{H} \cdot \mathbf{W} + \mathbf{b}$, the sigmoid function $\sigma(z)$ saturates to 1 as $z \to \infty$ and to 0 as $z \to -\infty$. The saturation points occur approximately at $z > \log(\frac{1}{\epsilon})$ and $z < -\log(\frac{1}{\epsilon})$, respectively.*

3. *The precision of the network's output is governed by the stable fixed points of the sigmoid function, which occur when $\sigma(z)$ stabilizes at values near 0 or 1. If the dynamics of the network converge to one or more stable fixed points, the effective precision is reduced because minor variations in the input will not significantly alter the output.*

4. *When three fixed points exist for the sigmoid function—two stable and one unstable—information encoding can become confined to the stable fixed points. This behavior causes the network to collapse to a discrete set of values, reducing its effective resolution.*

5. *Therefore, the maximum discrimination in the output is not only limited by $\epsilon$ but also by the attraction of the stable fixed points. The effective precision is bounded by both $1 - 2\epsilon$ and the dynamics that collapse the output towards these stable points.*

The detailed proof is discussed later in appendix.

**Theorem 14** *Consider a recurrent neural network (RNN) with fixed weights and the hidden state update rule given by:*

$$\mathbf{h}_{t+1} = \tanh(\mathbf{W}\mathbf{h}_t + \mathbf{U}\mathbf{x}_t + \mathbf{b}),$$

*where $\mathbf{W} \in \mathbb{R}^{d \times d}$, $\mathbf{U} \in \mathbb{R}^{d \times m}$, $\mathbf{b} \in \mathbb{R}^d$, and $\mathbf{x}_t$ represents the input symbol. Given a bounded sequence length $N$, the RNN can encode sequences of the form $a^n b^n c^n$ by exploiting state dynamics that converge to distinct, stable fixed points in the hidden state space for each symbol. The expressivity of the RNN, equivalent to a deterministic finite automaton (DFA), enables the encoding of such grammars purely through state dynamics.*

The detailed proof is discussed later in appendix.

**Theorem 15** *Given an RNN with fixed random weights and trainable sigmoid layer has sufficient capacity to encode complex grammars. Despite the randomness of the recurrent layer, the network can still classify sequences of the form $a^n b^n c^n$ by leveraging the distinct distributions of the hidden states induced by the input symbols. The classification layer learns to map the hidden states to the correct sequence class, even for bounded sequence lengths $N$.*

**Proof** The proof proceeds in three steps: (1) analyzing the hidden state dynamics in the presence of random fixed weights, (2) demonstrating that distinct classes (e.g., $a^n b^n c^n$) can still be linearly separable based on the hidden states, and (3) showing that the classification layer can be trained to distinguish these hidden state patterns.

**1. Hidden State Dynamics with Fixed Random Weights:**

Consider an RNN with hidden state $\mathbf{h}_t \in \mathbb{R}^d$ updated as:

$$\mathbf{h}_{t+1} = \tanh(\mathbf{W}\mathbf{h}_t + \mathbf{U}\mathbf{x}_t + \mathbf{b}),$$

where $\mathbf{W} \in \mathbb{R}^{d \times d}$ and $\mathbf{U} \in \mathbb{R}^{d \times m}$ are randomly initialized and fixed. The hidden state dynamics in this case are governed by the random projections imposed by $\mathbf{W}$ and $\mathbf{U}$.

Although the weights are random, the hidden state $\mathbf{h}_t$ still carries information about the input sequence. Specifically, different sequences (e.g., $a^n$, $b^n$, and $c^n$) induce distinct trajectories in the hidden state space. These trajectories are not arbitrary but depend on the input symbols, even under random weights.

**2. Distinguishability of Hidden States for Different Sequence Classes:**

Despite the randomness of the weights, the hidden state distributions for different sequences remain distinguishable. For example: - The hidden states after processing $a^n$ tend to cluster in a specific region of the state space, forming a characteristic distribution. - Similarly, the hidden states after processing $b^n$ and $c^n$ will occupy different regions.

These clusters may not correspond to single fixed points as in the trained RNN case, but they still form distinct, linearly separable patterns in the high-dimensional space.

**3. Training the Classification Layer:**

The classification layer is a fully connected layer that maps the final hidden state $\mathbf{h}_N$ to the output class (e.g., "class 1" for $a^n b^n c^n$). The classification layer is trained using a supervised learning approach, typically minimizing a cross-entropy loss.

Because the hidden states exhibit distinct distributions for different sequences, the classification layer can learn to separate these distributions. In high-dimensional spaces, even random projections (as induced by the random recurrent weights) create enough separation for the classification layer to distinguish between different classes.

Thus even with random fixed weights, the hidden state dynamics create distinguishable patterns for different input sequences. The classification layer, which is the only trained component, leverages these patterns to correctly classify sequences like $a^n b^n c^n$. This demonstrates that the RNN's expressivity remains sufficient for the classification task, despite the randomness in the recurrent layer. ∎

### E.1. Complete Proof of Precision Theorem

*Detailed proof of theorem 13*

**Proof** The proof proceeds in three steps: (1) analyzing the hidden state dynamics in the presence of random fixed weights, (2) demonstrating that distinct classes (e.g., $a^n b^n c^n$) can still be linearly separable based on the hidden states, and (3) showing that the classification layer can be trained to distinguish these hidden state patterns.

**1. Hidden State Dynamics with Fixed Random Weights:**

Consider an RNN with hidden state $\mathbf{h}_t \in \mathbb{R}^d$ updated as:

$$\mathbf{h}_{t+1} = \tanh(\mathbf{W}\mathbf{h}_t + \mathbf{U}\mathbf{x}_t + \mathbf{b}),$$

where $\mathbf{W} \in \mathbb{R}^{d \times d}$ and $\mathbf{U} \in \mathbb{R}^{d \times m}$ are randomly initialized and fixed. The hidden state dynamics in this case are governed by the random projections imposed by $\mathbf{W}$ and $\mathbf{U}$.

Although the weights are random, the hidden state $\mathbf{h}_t$ still carries information about the input sequence. Specifically, different sequences (e.g., $a^n$, $b^n$, and $c^n$) induce distinct trajectories in the hidden state space. These trajectories are not arbitrary but depend on the input symbols, even under random weights.

**2. Distinguishability of Hidden States for Different Sequence Classes:**

Despite the randomness of the weights, the hidden state distributions for different sequences remain distinguishable. For example: - The hidden states after processing $a^n$ tend to cluster in a specific region of the state space, forming a characteristic distribution. - Similarly, the hidden states after processing $b^n$ and $c^n$ will occupy different regions.

These clusters may not correspond to single fixed points as in the trained RNN case, but they still form distinct, linearly separable patterns in the high-dimensional space.

**3. Training the Classification Layer:**

The classification layer is a fully connected layer that maps the final hidden state $\mathbf{h}_N$ to the output class (e.g., "class 1" for $a^n b^n c^n$). The classification layer is trained using a supervised learning approach, typically minimizing a cross-entropy loss.

Because the hidden states exhibit distinct distributions for different sequences, the classification layer can learn to separate these distributions. In high-dimensional spaces, even random projections (as induced by the random recurrent weights) create enough separation for the classification layer to distinguish between different classes.

The main key insight observed based on above analysis is that even with random fixed weights, the hidden state dynamics create distinguishable patterns for different input sequences. The classification layer, which is the only trained component, leverages these patterns to correctly classify sequences like $a^n b^n c^n$. This demonstrates that the RNN's expressivity remains sufficient for the classification task, despite the randomness in the recurrent layer.

∎

*Detailed proof of theorem 14*

**Proof** The proof is divided into three parts: (1) establishing the existence of stable fixed points for each input symbol, (2) analyzing the convergence of state dynamics to these fixed points, and (3) demonstrating how the RNN encodes the sequence $a^n b^n c^n$ using these fixed points.

**1. Existence of Stable Fixed Points for Each Input Symbol:**
Let the hidden state $\mathbf{h}_t \in \mathbb{R}^d$ at time $t$ be updated according to:

$$\mathbf{h}_{t+1} = \tanh(\mathbf{W}\mathbf{h}_t + \mathbf{U}\mathbf{x}_t + \mathbf{b}),$$

where $\mathbf{x}_t \in \{a, b, c\}$ represents the input symbol. For a fixed input symbol $\mathbf{x}$, we analyze the fixed points of the hidden state dynamics.

The fixed points satisfy:

$$\mathbf{h}^* = \tanh(\mathbf{W}\mathbf{h}^* + \mathbf{U}\mathbf{x} + \mathbf{b}).$$

Assume that the system has distinct stable fixed points $\xi_a^-, \xi_b^-, \xi_c^-$ for inputs $\mathbf{x} = a$, $\mathbf{x} = b$, and $\mathbf{x} = c$, respectively. These fixed points are stable under small perturbations, meaning that for each symbol, the hidden state dynamics tend to converge to the corresponding fixed point.

**2. Convergence of State Dynamics to Fixed Points:**
For a sufficiently long subsequence of identical symbols, such as $a^n$, the hidden state will converge to $\mathbf{h} \approx \xi_a^-$ as $t$ increases. This convergence is governed by the stability of the fixed point $\xi_a^-$. The same holds true for subsequences $b^n$ and $c^n$, where the hidden state will converge to $\xi_b^-$ and $\xi_c^-$, respectively.

Mathematically, this convergence is characterized by the eigenvalues of the Jacobian matrix $\mathbf{J}$ at the fixed point $\xi_a^-$:

$$\mathbf{J} = \frac{\partial}{\partial \mathbf{h}}\left[\tanh(\mathbf{W}\mathbf{h} + \mathbf{U}\mathbf{a} + \mathbf{b})\right]\bigg|_{\mathbf{h}=\xi_a^-}.$$

If the eigenvalues satisfy $|\lambda_i| < 1$ for all $i$, the fixed point is stable, ensuring that the hidden state dynamics converge to $\xi_a^-$ over time.

**3. Encoding the Sequence $a^n b^n c^n$ via Fixed Points:**
Given a bounded sequence length $N$, the RNN can encode the sequence $a^n b^n c^n$ by leveraging the stable fixed points $\xi_a^-$, $\xi_b^-$, and $\xi_c^-$ as follows:

1. After processing the subsequence $a^n$, the hidden state converges to $\mathbf{h} \approx \xi_a^-$.

2. Upon receiving the input symbol $b$, the hidden state begins to transition from $\xi_a^-$ to $\xi_b^-$. As the network processes $b^n$, the hidden state stabilizes at $\xi_b^-$.

3. Similarly, the hidden state transitions to $\xi_c^-$ after processing $c^n$, representing the final part of the sequence.

■

### E.2. Estimation Methodology based on Machine Precision

Given the constraints of the RNN model and the precision limits of float32, we aim to calculate the maximum distinguishable count $N$ for each symbol in the sequence.

**Assumptions**

- The tanh activation function is used in the RNN, bounding the hidden state outputs within $(-1, 1)$.

- The machine epsilon ($\epsilon$) for float32 is approximately $1.19 \times 10^{-7}$, indicating the smallest representable change for values around 1.

- A conservative approach is adopted, considering a dynamic range of interest for tanh outputs from -0.9 to 0.9 to avoid saturation effects.

**Calculation Dynamic Range and Minimum Noticeable Change** The effective dynamic range for tanh outputs is set to avoid saturation, calculated as:

$$\text{Dynamic Range} = 0.9 - (-0.9) = 1.8.$$

Assuming a minimum noticeable change in the hidden state, given by $10 \times \epsilon$, to ensure distinguishability within the SGD training process, we have:

$$\Delta h_{\min} = 10 \times 1.19 \times 10^{-7}.$$

**Number of Distinguishable Steps** The total number of distinguishable steps within the dynamic range can be estimated as:

$$\text{Steps} = \frac{\text{Dynamic Range}}{\Delta h_{\min}}.$$

Given the usable capacity for encoding is potentially less than the total dynamic range due to the RNN's need to represent sequence information beyond mere counts, a conservative factor ($f$) is applied:

$$N_{\max} = f \times \text{Steps}.$$

**Conservative Factor and Final Estimation** Applying a conservative factor ($f$) to account for the practical limitations in encoding and sequence discrimination, we estimate $N_{\max}$ without dividing by 3, contrary to the previous incorrect interpretation. This factor reflects the assumption that not all distinguishable steps are equally usable for encoding sequences due to the complexity of sequential dependencies and the potential for error accumulation.

$$N_{\max} = f \times \frac{1.8}{10 \times 1.19 \times 10^{-7}}.$$

Thus we can shown that this estimation provides a mathematical framework for understanding the maximum count $N$ that can be distinguished by a simple RNN model with fixed weights and a trainable classification layer, under idealized assumptions about floating-point precision and the behavior of the tanh activation function. The actual capacity for sequence discrimination may vary based on the specifics of the network architecture, weight initialization, and training methodology.

