# OpenReview forum: "Bridging Neural and Symbolic Computation: A Learnability Study of RNNs on Counter and Dyck Languages"
_nesyconf.org/NeSy/2025/Conference_Phase_2 — NeSy 2025 - Phase 2 Poster_

### Official Review · Reviewer_kRWw · 2025-07-02
**A promising work that can be made more accessible**

**Rating:** 5
**Confidence:** 3

**Review:**

PROs:
- Mathematically rigorous
- Interesting research area

Suggestions:
- Unsure of the practical applications beyond the theoretical interest exposed by the problem at hand
- The neurosymbolic aspect of the solution itself is unclear or not well explicited
- Tables 1 and 2 seem to be "breakable" each into 2 more tables to better showcase results - most notably with respect to classifier-only and all layers)
- Figure 4: How to interpret that LSMT performs better than O2RNN with uniform initialization? Also Figure 4 is hard to read or seems misrepresented ((a) doesn't seem to refer to a plot)
- The body of the paper does not mention some of the Appendixes, most notably Appendix A, which makes the paper very hard to read without.
- The structure of the paper is to be reconsidered for easier reading (e.g. including the definitions and examples of the languages used for clarity in the body itself)

VERDICT: A promising work that may benefit from some polishing by addressing key points to render it more accessible and appreciable by the interested audience.

**Anonymity:**

Remain anonymous

---

### Official Review · Reviewer_aFHU · 2025-07-06
**Interesting analysis on the discrepancy between RNN's expressivity and learnability.**

**Rating:** 6
**Confidence:** 3

**Review:**

* Summary

This paper provides a study of the discrepancy between theoretical
expressivity and learnability of Recurrent Neural Networks on two
types of languages (Counter and Dyck). The paper is well written, the
research question is clear with an exhaustive evaluation.


* Comments

  - This paper introduces three types of sampling negatives which differ
  on the similarity to the associated positive sample. I wonder what
  the positive:negative ratio is.  Since the data is
  inherently sparse, I wonder if having a large number of negatives
  could impact the performance of the models.

  - Although the evaluation is exhaustive, it uses small neural networks
  (2-12 neurons). I understand that theoretically, small networks
  should be able to handle the languages you explore. However, it is
  also mentioned that "architectural modifications can significantly
  alter the network's functional capacity". It would be valuable to see
  if these findings hold with larger network sizes as well.

  - In the Related Work section, authors state that RNNs with ReLU
  activation has shown better performance (in counting tasks) than
  sigmoid or tanh. However, in following sections authors
  decide to analyze only sigmoid and tanh. This is a bit intriguing
  and I would suggest elaborating on the rationale of this decision
  because I might be missing something here.

  - The paper has good diagnostic points. I would suggest adding some
  discussion on what potential paths can be followed to overcome the
  limitations that the paper establishes.

  - "The proofs for the theorem and the above corollaries is provided in
  the Appendix" --> Please add which Appendix: A,B,C,....

  - In Tables 1,2. Would you please state which metric is being
  evaluated? Is the metric defined somewhere? Also, I think it needs
  some reformatting (i.e., "8.5" --> "8.50")


* Minor comments

  - Found both "counter" and "Counter". Also "dyck" and "Dyck". I
  suggest to standardize the notation.
  - "The proofs for the theorem and the above corollaries is provided" --> "The proofs for the theorem and the above corollaries are provided"

**Anonymity:**

Remain anonymous

---

### Official Review · Reviewer_rUUC · 2025-07-08
**Interesting, recommend acceptance**

**Rating:** 8
**Confidence:** 2

**Review:**

This paper presents a neuro-symbolic analysis of the practical learnability of Recurrent Neural Networks (RNNs), specifically LSTMs and O2RNNs. The authors challenge the focus on theoretical expressivity by investigating how these models perform on structured formal languages -- counter and Dyck languages -- under realistic training constraints. The core finding is a significant gap between what these architectures are theoretically capable of doing, i.e. operating as stack-based automata, and what they empirically learn, i.e. operating as simpler finite-state machines. This work introduces a mathematical framework based on the fixed-point theory to analyse the stability of activation functions, connecting the theoretical properties of the neural network to its empirical performance.

The paper is well written, and the experimental findings provide an interesting contribution to a deeper understanding of the learnability of LSTMs and other recurrent neural networks. However, some concepts (e.g. O2RNN, Dyck languages) could be presented in a clearer and more in-depth way to facilitate the reader and make the paper more self-contained. In fact, the "Background & Methodology" section is missing.

Here are some comments and suggestions:
1. In Sec. 4 at "Models", link the references to the sentence, e.g.:
    - "Long Short-Term Memory (LSTM) networks by Hochreiter and Schmidhuber (1997)"
    - "Second-Order Recurrent Neural Networks (O2RNNs) by Omlin and Giles (1992a)"
2. In Sec. 5 at "Stability of Second-Order RNNs", it should be more correct to say "less than 4%" according to the results in the tables, right?
3. In Sec. 5 at "Effect of Initialization on Fixed-Point Stability", it should be "...declines from hard 0 to hard 2..." according to Fig. 4.

**Anonymity:**

Remain anonymous